# Development Trajectories of Fatigue, Quality of Life, and the Ability to Work among Colorectal Cancer Patients in the First Year after Rehabilitation—First Results of the MIRANDA Study

**DOI:** 10.3390/cancers15123168

**Published:** 2023-06-13

**Authors:** Tomislav Vlaski, Marija Slavic, Reiner Caspari, Harald Fischer, Hermann Brenner, Ben Schöttker

**Affiliations:** 1Division of Clinical Epidemiology and Aging Research, German Cancer Research Center (DKFZ), 69120 Heidelberg, Germany; 2Medical Faculty Heidelberg, Heidelberg University, 69120 Heidelberg, Germany; 3Clinic Niederrhein, 53474 Bad Neuenahr-Ahrweiler, Germany; 4Clinic Rosenberg, 33014 Bad Driburg, Germany; 5Division of Preventive Oncology, German Cancer Research Center (DKFZ), 69120 Heidelberg, Germany; 6National Center for Tumor Diseases (NCT), German Cancer Consortium (DKTK), German Cancer Research Center (DKFZ), 69120 Heidelberg, Germany

**Keywords:** colorectal cancer, rehabilitation, fatigue, ability to work, quality of life

## Abstract

**Simple Summary:**

Colorectal cancer (CRC) patients often experience fatigue, poor ability to work, and low quality of life (QoL) after therapy. We conducted a prospective study of 147 CRC patients who underwent a three-week in-patient rehabilitation clinic visit in Germany. Patients completed questionnaires at the start of rehabilitation and at regular 3-month intervals for up to a year. We found a strong correlation between fatigue and QoL, and moderate correlations between fatigue and the ability to work, as well as between QoL and the ability to work. Fatigue, QoL, and ability to work improved significantly from the start of rehabilitation to the three-month follow-up, and there was little change afterward in the first year after rehabilitation. In summary, fatigue, QoL, and the ability to work were highly interconnected in CRC patients, and all of them improved from the start of rehabilitation to the 3-month follow-up.

**Abstract:**

Cancer-related fatigue, low quality of life (QoL), and low ability to work are highly prevalent among colorectal cancer (CRC) patients after tumor surgery. We aimed to analyze their intercorrelations and trajectories in the first year after in-patient rehabilitation in the German multicenter MIRANDA cohort study. Recruitment is ongoing, and we included the first 147 CRC patients in this analysis. Participants filled out questionnaires at the beginning of in-patient rehabilitation (baseline) and at 3, 6, 9, and 12 months after the baseline. The EORTC-QLQ-C30-General-Health-Status (GHS)/QoL, the FACIT-F-Fatigue Scale, and the FACIT-F-FWB-ability-to-work items were used to evaluate QoL, fatigue, and ability to work, respectively. The fatigue and QoL scales were highly correlated (r = 0.606). A moderate correlation was observed between the fatigue and ability to work scales (r = 0.487) and between the QoL and ability to work scales (r = 0.455). Compared to the baseline, a statistically significant improvement in the QoL, ability to work, and fatigue scales were observed at the 3-month follow-up (Wilcoxson signed rank test, all *p* < 0.0001). The three scales plateaued afterward until the 12-month follow-up. In conclusion, fatigue, QoL, and ability to work were highly interrelated, improved quickly during/after in-patient rehabilitation, and did not change much afterward in German CRC patients.

## 1. Introduction

Colorectal cancer (CRC) presents a significant burden to global health, with almost 2 million new cases and 1 million deaths worldwide in 2020, ranking as the third and second most frequent cause of cancer incidence and mortality, respectively [1]. In Germany, about one of eight newly diagnosed cancers affect the colon or rectum. In 2019, approximately 26,270 women and 32,701 men were newly diagnosed [2]. Current incidence rates indicate that one in nineteen women and one in sixteen men will be diagnosed with CRC during their lifetime. About two-thirds of these cancers will be detected in the colon. Prognosis still depends mainly on the disease stage at diagnosis, with relative five-year survival rates of about 90% for stage I, about 70% for stage III patients, and about 10% for stage IV patients [3]. 

In Germany, around two out of three CRC patients undergo oncological rehabilitation [4], which is an integral part of the German healthcare system and a critical component of modern cancer treatment that usually immediately follows operation, chemo-, or radiotherapy. Additionally, it can be used to address functional disabilities that persist years after cancer treatment. Multidisciplinary interventions that combine physical, psycho-educational, and vocational components are the basis of oncologic rehabilitation programs. Patients usually spend three weeks at a specialized rehabilitation clinic and the costs are covered by the German Pension Fund with a small co-payment per day by the patient. Over three weeks, patients undergo comprehensive and progressive practical training and a theoretical education program to acquaint them with essential aspects to be implemented independently at home. This includes enhancing physical activity, the self-managing of joint, respiratory, and incontinence issues, and the self-administration of stoma care. Patients learn to adopt a healthier diet and optimize food consumption, particularly after surgeries involving the stomach, pancreas, or esophagus. Therapeutic approaches addressing chemotherapy-induced polyneuropathy (CIPN) can also be taught, allowing patients to apply these techniques independently. Mental health concerns, such as sleep disturbances, depression, or anxiety, can be tackled by psycho-oncology counseling services and by learning relaxation methods or mindfulness practices. Furthermore, initiatives aimed at reintegrating patients into professional and social settings are integral components of oncological rehabilitation (e.g., through phased reintegration and prescription of assistive devices and tools) [5]. 

Studies have shown that more than half of all cancer survivors require an absence from work while receiving cancer therapy and coping with the associated disability. Still, most cancer patients return to work after treatment [6]. However, cancer patients were still found to have a higher risk of job loss, a lower probability of re-employment, and a prolonged absence from work than patients with other diseases treated in rehabilitation clinics [6]. Furthermore, unemployment among cancer survivors adversely affected their QoL [7]. In addition, a reduced household income, declined physical abilities, and psychosocial repercussions were shown to influence the prognosis of underlying diseases, such as cancer [8]. Regarding reintegration into the labor market of colorectal cancer patients in Germany, a recent study showed that 32.5% of CRC survivors of working age < 60 years did not return to work within ten years [4]. 

Cancer-related fatigue is one of the main risk factors for a disability pension because it can severely limit the ability to work [9]. In a study of cancer survivors, 91% of patients with fatigue reported moderate to severe functional impairment, while only 30% of those not having fatigue reported these impairments [10]. In addition, fatigue is a significant cause of reduced QoL in CRC patients, with more than one-third of long-term survivors reporting a low QoL after treatment [11]. 

The goal of in-patient rehabilitation of CRC patients in Germany is to improve their general health status and, subsequently, their ability to work and QoL. Cancer-related fatigue is a main challenge to achieving this goal. The first aim of this study was to assess the frequencies, extent, and correlations of fatigue, impairments in QoL, and the subjective ability to work in a cohort of CRC patients recruited during in-patient rehabilitation. The second aim was to show the trajectories of fatigue, QoL, and the ability to work from the start of in-patient rehabilitation over 12 months. 

## 2. Materials and Methods

### 2.1. Study Design and Participants

The MIRANDA study is a multicenter, prospective cohort study with the aim to investigate risk, preventive, and prognostic factors of fatigue, QoL, the ability to work, and other health outcomes among CRC patients after in-patient rehabilitation. The cohorts’ participants are being recruited in 6 rehabilitation clinics distributed all over Germany to be representative of the German rehabilitation setting. However, the patients for this interim analysis were mainly recruited from 2 rehabilitation clinics in North Rhine-Westphalia and Rheinland-Pfalz. The following inclusion criteria were applied: Age ≥ 18 years.CRC diagnosis (ICD-10 C18-20 and C21.8) and recruitment within 12 months after primary CRC therapy (surgery, radiation, and/or chemotherapy).At least three weeks of in-patient rehabilitation in one of the six cooperating rehabilitation clinics.Sufficient knowledge of the German language and mental capability to give written informed consent and comply with the study requirements.

No written informed consent was the only exclusion criterion. The approval of the Ethical Committee of the Faculty of Medicine Heidelberg was received in January 2020 (S-905/2019). The MIRANDA study was registered ahead of the start of recruitment in the German Clinical Trials Register, No. DRKS00020822.

Recruitment started in Sept 2020 and will last at least until March 2025, with the aim to recruit approximately 1000 study participants. The included patient population will be followed for a maximum of 10 years. 

At the baseline (the first week during in-patient rehabilitation), study participants fill out a comprehensive questionnaire and donate blood, urine, and stool samples. DNA is extracted, and a large set of commonly measured biomarkers are analyzed immediately (e.g., blood cell counts, C-reactive protein, HbA_1c_, lipids, and 25-hydroxyvitamin D).

The MIRANDA study participants with 25-hydroxyvitamin D < 60 nmol/L and no contraindication for vitamin D supplementation are invited to additionally take part in the “*Personalized vitamin D supplementation for reducing or preventing fatigue and enhancing the quality of life of patients with colorectal tumor-randomized intervention trial*” (VICTORIA) trial [12,13]. Approximately half of the MIRANDA participants also participate in VICTORIA and are randomly assigned to receive either vitamin D_3_ or a placebo for 12 weeks.

Every 3 months in the first year and then after 3, 5, 7, and 10 years, study participants are again asked to fill out self-administered questionnaires containing the following validated endpoint scales: European Organization for the Research and Treatment of Cancer—Quality of Life Questionnaire—Core 30 (EORTC-QLQ-C30) [14];EORTC-QLQ ColoRectal cancer-specific QoL (EORTC-QLQ-CR29) [15];EORTC-QLQ Fatigue Assessment (EORTC-QLQ-FA12) [16];Functional Assessment of Chronic Illness Therapy—Fatigue—Fatigue Scale (FACIT-F-FS) [17];FACIT-F—Functional Well-Being (FACIT-F-FWB) scale [17];Questionnaire on Stress in Cancer (QSC-R10) [18];Geriatric Depression Scale (GDS-15) [19];General Anxiety Disorder-7 (GAD-7) scale [20];Fatigue, Resistance, Ambulation, Illnesses, and Loss of Weight (FRAIL) scale [21,22].

Furthermore, return to work is an important study endpoint defined by self-reported paid work of ≥3 h/week.

### 2.2. Outcomes of Interest in This Work

#### 2.2.1. Quality of Life

Global QoL was determined using the global health status (GHS)/QoL scale of the EORTC-QLQ-C30, version 3.0. The final score for each participant is calculated and converted into percentage points (0 to 100). Higher scores represent better functioning. A low-to-moderate QoL was defined by an EORTC-QLQ-C30-GHS/QoL score of <70 points [23]. A mean difference ≥ 5 points in the GHS/QoL scale is considered a clinically relevant improvement in QoL [24].

Furthermore, the EORTC-QLQ-C30 function and symptom sub-scales were of interest in this work, and they are also expressed in percentage point scales from 0 to 100. 

#### 2.2.2. Fatigue

Fatigue was evaluated using the FACIT-F-FS, version 4.0. The tool assesses self-reported tiredness, weakness, and difficulty conducting everyday activities due to fatigue. The total maximum score is 52 points. Higher scores represent less fatigue. Cancer-related fatigue was assumed to be present for study participants with a FACIT-F-FS score < 34 points [25]. A mean difference of ≥3 FACIT-F-FS points is considered a clinically relevant improvement in fatigue symptoms [26]. 

#### 2.2.3. Ability to Work

Subjective ability to work was determined by the FACIT-F-FWB-Ability to Work (AW) item (“I am able to work, including work at home”). Responses for the FACIT-F-FWB-AW are configured on a scale from 0 to 4 points: “Not at all” (0 points), “A little bit” (1 point), “Somewhat” (2 points), “Quite a bit” (3 points), and “Very much” (4 points). Low-to-moderate ability to work was defined by the response categories “Not at all” and “A little bit”.

In contrast to the gold standard assessment method, the Work Ability Index (WAI) [27], the FACIT-F-FWB-AW can be answered by both study participants who currently work and do not work (e.g., due to sick leave, retirement, or unemployment) and thus could be answered by all MIRANDA study participants at all time points. The WAI is a summary measure of seven items (range 7–49), from which an index is calculated: 1 = “critical”, 2 = “moderate”, 3 = “good”, and 4 = “very good”. The validated short-form of the WAI was only applied in the 9-month follow-up and was used to validate the FACIT-F-FWB-AW among study participants who worked at that time.

The four categories of the WAI were reduced to three by merging the “good” and “very good” categories. The five categories of the FACIT F-FWB-AW were also reduced to three by combining the lower three answers (“Not at all”, “A little”, and “Somewhat”) into one. The cross-tabulation of these categories for the two questionnaire instruments is shown for *n* = 49 study participants who worked at the 9-month follow-up in Appendix A. With a weighted kappa statistic of 0.29 (95% confidence intervals (95% CI): 0.08–0.50), the agreement between both instruments was in the “fair agreement” range (0.21–0.40) of the weighted kappa statistic [28]. 

### 2.3. Statistical Analysis

The current investigation includes only those 147 study participants recruited between 09/2020 and 12/2021 who had the chance to have completed the 12-month follow-up at the time of analysis, which took place in 02/2023. 

Means with 95% CIs were estimated for the EORTC-QLQ-C30-GHS/QoL, the FACIT-F-FS, and the FACIT-F-FWB-AW at the baseline and 3-, 6-, 9-, and 12-month follow-up and were displayed graphically for the total study population, as well as distinctly by sex, age (<65/≥65 years), cancer stage (I–III), and participation in the VICTORIA trial. The Wilcoxon signed-rank test was used to assess potential statistically significant differences between responses at follow-ups and the baseline. Furthermore, means with 95% CIs were estimated for the EORTC-QLQ-C30 function and symptom sub-scales in the total population at the baseline and 3-, 6-, 9-, and 12-month follow-up are shown graphically. 

The Spearman rank test was used to estimate correlation coefficients between the EORTC-QLQ-C30-GHS/QoL, FACIT-F-FS, and FACIT-F-FWB-AW at the baseline and their changes from the baseline to the 3-month follow-up. For the dichotomized variables of these scales, cancer-related fatigue, low-to-moderate QoL, and low-to-moderate ability to work, a proportionate Venn diagram was used to present their co-occurrence graphically. In addition, logistic regression models adjusted for age, sex, cancer stage, radiation, chemotherapy, and time since surgery were used to assess odds ratios (ORs) and 95% CIs for the associations of (a) fatigue and low-to-moderate QoL, (b) fatigue and low-to-moderate ability to work, and (c) low-to-moderate QoL and low-to-moderate ability to work.

Participants with missing data were excluded from respective analyses. All analyses were performed using SAS software version 9.4, and all tests were two-sided and considered statistically significant for *p*-values < 0.05.

## 3. Results

### 3.1. Partcipants’ Characteristics

The baseline characteristics of the 147 enrolled study participants can be found in Table 1. The mean age of all participants was 62.5 years, 37.4% were female, and CRC stages I-III were approximately equally distributed. All study participants had CRC surgery before enrollment and 45.5% were admitted to a rehabilitation clinic in the first month after surgery. Almost half of the participants were fully employed (47.5%), about one-third of participants were retired (39.7%), and the rest of the participants were either part-time employed (9.2%) or unemployed (3.5%).

### 3.2. Correlation of Fatigue, Quality of Life, and Ability to Work Scales at the Baseline

A strong correlation was found between the FACIT-F-FS and the EORTC-QLQ-C30-GHS/QoL at the baseline (Spearman correlation coefficient (r_s_) = 0.606; *p* < 0.0001)). The correlation between the FACIT-F-FS and the FACIT-F-FWB-AW was moderate (r_s_ = 0.487, *p* < 0.0001). Furthermore, a moderate correlation was observed between the EORTC-QLQ-C30-GHS/QoL and the FACIT-F-FWB-AW (r_s_ = 0.455, *p* < 0.0001). 

### 3.3. Co-Occurrence of Fatigue, Low-to-Moderate Quality of Life, and Low-to-Moderate Ability to Work

The prevalence of cancer-related fatigue, defined by a FACIT-F-FS score < 34 points, was 55% (*n* = 80/145; 2 persons with missing values). The prevalence of low-to-moderate QoL, defined by an EORTC-QLQ-C30-GHS/QoL score < 70 points, was 83% (*n* = 122/147). The number of participants with a low-to-moderate ability to work, defined as either “a little” or “not at all” in the FACIT-F-FWB-AW score, was 42% (*n* = 61/145; 2 persons with missing values). 

Overall, 144 study participants had complete data on the three scales at the baseline, and only a minority of 9% (*n* = 13/144) had neither fatigue, low-to-moderate QoL, nor low-to-moderate fatigue. A Venn diagram illustrating the overlap of cancer-related fatigue, low-to-moderate ability to work, and low-to-moderate QoL among *n* = 131 study participants who had at least one of the three is shown in Figure 1. The prevalence of individuals who scored low on all three scales was the highest, at 35% (*n* = 46/131). The second largest group was participants with only a low-to-moderate QoL (30%; *n* = 39/131), and the third largest group included participants with both cancer-related fatigue and a low-to-moderate ability to work (*n* = 21%; *n* = 28/131). Other combinations or individuals who had only cancer-related fatigue (2%; *n* = 3/131) or only a low-to-moderate ability to work (2%; *n* = 3/131) were rather rare.

### 3.4. Adjusted Odds Ratios for the Cross-Sectional Associations of Cancer-Related Fatigue, Low-to-Moderate Quality of Life, and Low-to-Moderate Ability to Work at the Baseline

At the baseline, when adjusted for age, sex, cancer stage, chemotherapy, radiation, and time since surgery, the odds of patients with fatigue having a low-to-moderate QoL increased by more than seven times (OR [95% CI]: 7.77 [2.40–25.18]), and the odds of having a low-to-moderate ability to work increased by more than twelve times (ORs [95% CI]: 12.30 [4.80–31.49]). The association of a low-to-moderate QoL and a low-to-moderate ability to work was weaker and not statistically significant (OR [95% CI]: 2.71 (0.96–7.67)). 

A longitudinal analysis was not conducted because the overlap of subjects with fatigue and a low-to-moderate QoL or ability to work at the baseline was too high, and they could not be adjusted for each other due to high multicollinearity. 

### 3.5. Development Trajectories of Fatigue, Quality of Life, and Ability to Work Scales

The response rates at the follow-ups were high, with *n* = 119 (81.0%) participants in the 3- and 6-month follow-up and *n* = 114 (77.6%) participants in the 9- and 12-month follow-up. Overall, *n* = 114 study participants (77.6%) filled out all questionnaires and were used for this analysis on development trajectories. Non-response to follow-up questionnaires was not related to the most important baseline characteristics of age, sex, cancer stage, and additional participation in the VICTORIA study (Appendix A).

The mean (95% CI) of the FACIT-F-FS score was 30.8 points (28.6–32.9) at the baseline. The mean improved statistically significantly by 7.3 points to 38.1 points (36.0–40.2) until the 3-month follow-up, which was more than twice the minimally clinically relevant difference of 3 FACIT-F-FS score points (Figure 2A). With increasing follow-up time, the changes in the FACIT-F-FS were small and not clinically relevant. The mean FACIT-F-FS scores were 38.9 (37.0–40.8), 38.8 (36.8–40.9), and 38.6 (36.7–40.5) points at the 6-, 9-, and 12-month follow-ups, respectively.

The mean (95% CI) of the EORTC-QLQ-C30-GHS/QoL score was 49.2 points (45.6–52.9), and it significantly (*p* < 0.001) improved by 14.2 points until the 3-month follow-up (63.4 (59.8–67.1) points; Figure 2B). This is more than twice the minimally relevant difference of five score points. The mean EORTC-QLQ-C30-GHS/QoL reached a plateau at the 3-month follow-up, and the mean values did not change much afterward, with values of 62.6 (58.26–67.2), 63.4 (58.6–66.2), and 63.4 (59.6–67.2) points at the 6-, 9-, and 12-month follow-up, respectively.

The mean (95% CI) of the FACIT-F-FWB-AW score was 1.7 (1.5–1.9) points at the baseline. It improved significantly (*p* < 0.001) by 0.8 points until the 3-month follow-up (2.5 (2.3–2.8) points) and remained stable afterward, with values of 2.7 (2.5–2.9), 2.5 (2.3–2.8), and 2.7 (2.5–2.9) points at the 6-, 9-, and 12-month follow-up, respectively (Figure 2C).

The development trajectories of fatigue, QoL, and the ability to work, stratified by sex, age, stage, participation in the VICTORIA trial, and time since surgery, are shown in Appendix A, respectively. No relevant differences in the trajectories were observed in the subgroup analyses. It is noteworthy, however, that men had higher FACIT-F-FS scores (i.e., less fatigue) than women at all time points, and the development trajectory curves were almost parallel (Appendix A).

### 3.6. Correlation of Changes in Fatigue, Quality of Life, and Ability to Work Scales from the Baseline to 3-Month Follow-up

As clinically relevant changes in the scales were only observed from the baseline to 3-month follow-up, only the correlations in this period were analyzed. A moderate correlation was found between the changes in the FACIT-F-FS and EORTC-QLQ-C30-GHS/QoL scales from the baseline to 3-month follow-up (r_s_ = 0.598; *p* < 0.0001). The correlation between the changes in the FACIT-F-FS and the FACIT-F-FWB-AW was weak (r_s_ = 0.347, *p* < 0.0001). Furthermore, a weak correlation was observed between changes in the EORTC-QLQ-C30-GHS/QoL and FACIT-F-FWB-AW (r_s_ = 0.356, *p* < 0.0001).

### 3.7. Development Trajectories of the EORTC-QLQ-C30’s Functional Scales

The functional scales of the EORTC-QLQ-C30 had the same development trajectory as the EORTC-QLQ-C30-GHS/QoL, FACIT-F-FS, and FACIT-F-FWB-AW, with a statistically significant improvement between the baseline and the 3-month follow-up and no further relevant changes afterward (Figure 3). The largest improvement was observed in the role functioning scale, which improved from 36.9 (31.1–42.8) points at the baseline by 27.7 points until the 3-month follow-up. The smallest improvement was observed in the cognitive functioning scale, which improved from 73.1 (67.9–78.3) points at the baseline by 5 points until the 3-month follow-up. However, all comparisons of follow-up scale values with the baseline values were statistically significant.

### 3.8. Development Trajectories of the EORTC-QLQ-C30’s Symptom Scales

The symptom scales of the EORTC-QLQ-C30 are shown in Figure 4. Five symptom scales show very similar trajectories to the EORTC-QLQ-C30-GHS/QoL, FACIT-F-FS, and FACIT-F-FWB-AW with statistically significant improvement until the 3-month follow-up and a plateau afterward. These were the fatigue, pain, dyspnea, insomnia, and appetite loss scales. The other four scales (nausea and vomiting, constipation, diarrhea, and financial difficulties) did not change relevantly during follow-up.

## 4. Discussion

Our study showed that fatigue, QoL, and the ability to work were highly correlated in CRC patients undergoing in-patient rehabilitation. Trajectories of fatigue, QoL, and self-perceived ability to work showed very similar trends over time. The most significant improvements were observed in the first three months after the start of in-patient rehabilitation. These improvements remained constant thereafter for at least one year after rehabilitation. The functional QoL scales showed the same development trajectories, with the largest improvement in role functioning. Furthermore, five out of the nine symptom scales improved with a very similar development trajectory (fatigue, pain, dyspnea, insomnia, and appetite loss), whereas other symptom scales showed little changes over time (nausea/vomiting, constipation, diarrhea, and financial difficulties). 

Previous studies have demonstrated that fatigue in cancer patients is usually presented with other symptoms and conditions, such as pain and depression [34]. Because of its pervasive qualities, fatigue often leads to a disruption of everyday social activities and a decline in cognitive and physical functioning [35]. Patients can experience this as a negative impact on their QoL, while the lack of energy can make them less functional and subsequently reduce their ability to work [36]. Previous studies demonstrated that the perceived ability to work is inversely associated with levels of fatigue in cancer survivors [36]. 

Our study found a strong correlation between cancer-related fatigue and low QoL and a moderate correlation between fatigue and a low ability to work among CRC patients, which supports observations in previous studies with other cancer types. In CRC patients, the impact of invasive surgical procedures and chemotherapy on fatigue symptoms is particularly emphasized [37,38]. After adjusting for age, sex, and these cancer-related factors, strong associations between fatigue and QoL (OR > 7), as well as between fatigue and the ability to work (OR > 12), persisted. Taken together, our findings suggest that fatigue, low QoL, and low ability to work are interrelated and clustered in CRC patients after primary treatment.

In our study, fatigue, QoL, and the ability to work improved in the first 3 months after the start of in-patient rehabilitation and did not change much in the following 9 months.

The time following cancer treatment, such as surgery or chemotherapy, has a major impact on fatigue levels and, subsequently, QoL and the ability to work in cancer patients [37,39]. Interestingly, the trajectories did not differ among study participants who were enrolled up to 1 month after CRC surgery and those with surgery 2–12 months before rehabilitation. Furthermore, the baseline fatigue levels of these two groups showed no differences. Most of the study participants had received chemotherapy already before in-patient rehabilitation (53% of the total study population). The proportions of study participants who received chemotherapy or radiotherapy during or at 3, 6, 9, and 12 months after rehabilitation, were low, with values of 2%, 7.5%, 7.5%, 7.5%, and 9.5%, respectively. This relatively low and stable proportion of subjects with chemotherapy during follow-up might explain the plateau of relatively low fatigue, QoL, and the ability to work between the 3- and 12-month follow-up. 

The subgroup analyses by age showed that patients younger than 65 years had lower QoL, fatigue, and the ability to work scores at the baseline than older patients, but the difference was not statistically significant in our rather small study. However, the trajectories of QoL, fatigue, and the ability to work in the following 12 months did not differ by age. The initially lower baseline QoL score in younger compared to older CRC patients was also demonstrated in a previous study with CRC patients, which compared it to the QoL of a population-based cohort [40]. Only the QoL of the younger CRC patients was relevantly lower than the reference population. This might be explained by differences in how younger and older adults perceive the detrimental effects of disease and their symptoms. Older patients may rate these adverse effects as appropriate for their age.

Our study design cannot answer the question of whether the rehabilitation program caused fatigue improvement because we did not have a control group that did not have in-patient rehabilitation. In our study, patients took part in a specialized rehabilitation program, which is based on the WHO-recommended “bio-psycho-social model,” comprising physical therapy, exercise, psychological training and assistance, and social participation [41]. It is possible that this type of multimodal approach could be responsible for the observed reduction in fatigue because physical activity and cognitive behavioral therapy interventions have been shown to effectively reduce fatigue in CRC patients [42,43,44]. However, it is just as possible that the same fatigue improvement could have been observed without in-patient rehabilitation because some studies showed that fatigue levels could reduce after treatment, even without any intervention or medication, possibly due to the gradual reduction in inflammation caused by treatment [45]. 

It is evident that certain patients are more prone to having cancer-related fatigue, and there are several risk factors already identified, such as late-stage disease, female sex, younger age, and surgery or chemotherapy [46,47]. These factors should be considered during rehabilitation to identify patients who might need additional support. Accordingly, we observed higher fatigue levels in women with CRC in our study, which improved until the 3-month follow-up but never caught up to men in the first year after rehabilitation. 

The QoL of CRC patients in our study improved significantly in the first 3 months since the start of rehabilitation and stayed almost unchanged in study months 4–12. This is in agreement with a study in Austria, which demonstrated that QoL increased during and after rehabilitation, and symptoms diminished [48]. However, different trajectories of QoL were reported for CRC patients who did not participate in a rehabilitation program. A systematic review found that QoL usually decreases up to 6 months after treatment and then increases to pre-treatment levels within one year [49]. It was suggested that these detriments of QoL in the first six months are due to post-treatment effects that can last several months, including fatigue, loss of appetite, and anxiety [50]. Maybe, the timing of the chemotherapy differed in the study populations who had rehabilitation and who did not, which might explain the different QoL trajectories.

The ability to work and the rate of return to work have emerged as crucial indicators of successful rehabilitation, given that the ultimate goal of rehabilitation is to restore an individual’s health and functional status to a degree as close as possible to their pre-illness condition. Cancer patients who retain employment after therapy have a higher QoL, experience fewer symptoms, and have lower fatigue [51]. However, the chain of events could also be reversed, i.e., cancer patients with lower fatigue have a higher QoL and are more likely to retain employment after therapy. A Dutch study, which evaluated the ability to work of CRC patients for two years after treatment found that the ability to work decreased in the first six months after treatment, followed by an increase to near-baseline values from 6 to 24 months [52,53]. Maybe, the different trajectory in our study can again be explained by the fact that surgery and chemotherapy for most study participants already took place up to 12 months prior to enrollment in our study. 

The MIRANDA study has several strengths, including its multicenter and prospective design. By recruiting patients with minimal in/exclusion criteria from six rehabilitation clinics in Germany and following them for up to 10 years, the study establishes a cohort representative of the German rehabilitation setting with long-term follow-up. The use of validated questionnaire instruments ensures the reliability and validity of the collected data and allows for direct comparisons with other studies that employ the same instruments.

The main limitation of this preliminary analysis of the MIRANDA study is its low sample size (*n* = 147), which limits the statistical power of our analysis. The statistical power should increase substantially when the MIRANDA study has reached its goal of 1000 study participants. Furthermore, it should be noted that the patients included in the present preliminary analyses were recruited during the height of the COVID-19 pandemic from September 2020 to end of December 2021. During this time, occupancy of the participating clinics was reduced to approx. 20–50% of the usual pre-pandemic occupancy. This particular context may have introduced a specific selection of patients. Therefore, it is possible that updated results from the MIRANDA study with the final study population of *n* = 1000 CRC patients may not align with the current findings with the first recruited *n* = 147 patients.

Finally, we would like to state that our results may not be generalizable to patient populations from other countries with different (or even no) rehabilitation programs.

## 5. Conclusions

In summary, our study found that fatigue, QoL, and the ability to work were interconnected and commonly experienced by CRC patients in Germany up to 12 months after surgery. Fatigue, QoL, and the ability to work improved quickly during and after in-patient rehabilitation and did not change much afterward.

## Figures and Tables

**Figure 1 cancers-15-03168-f001:**
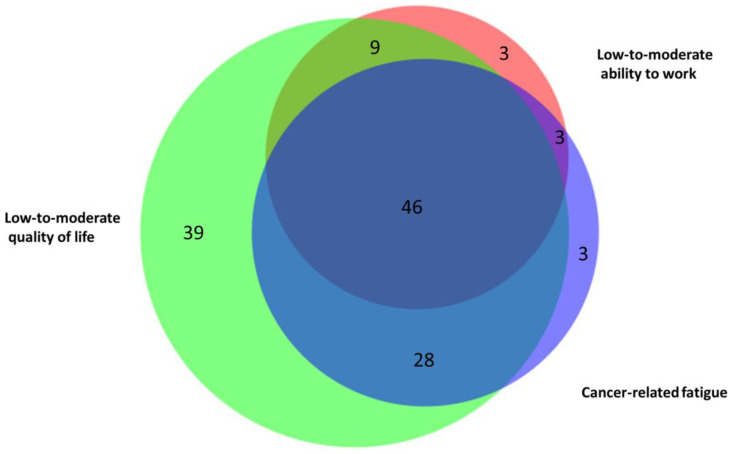
Venn diagram of *n* = 131 participants with cancer-related fatigue (purple), a low-to-moderate ability to work (red), and/or a low-to-moderate quality of life (green).

**Figure 2 cancers-15-03168-f002:**
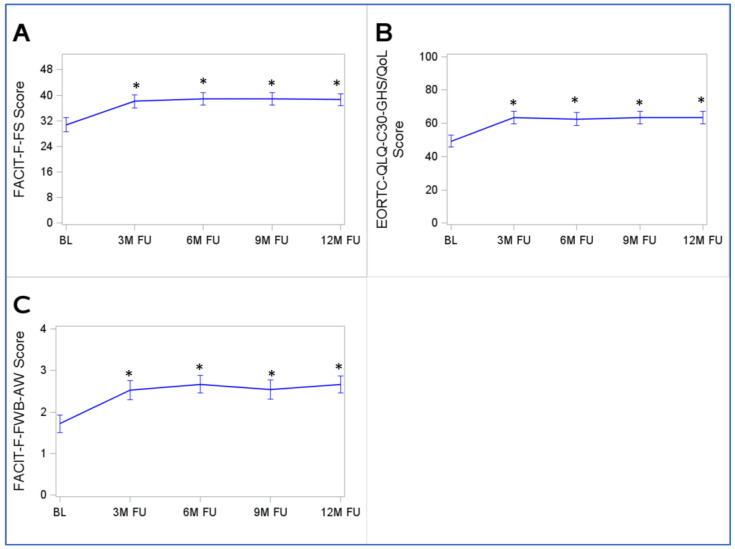
Trajectories of the FACIT-F-FS (**A**), EORTC-QLQ-C30-GHS/QoL (**B**), and FACIT-F-FWB-AW (**C**) from the baseline to the 12-month follow-up. Abbreviations: BL, baseline; 3M FU, 3-month follow-up; 6M FU, 6-month follow-up; 9M FU, 9-month follow-up; 12M FU, 12-month follow-up; * *p* < 0.05 of the Wilcoxon signed-rank test comparing responses at follow-ups with those at the baseline. Note: the figure displays means with 95% confidence intervals. Only study participants with complete follow-up data were included in this analysis (*n* = 114).

**Figure 3 cancers-15-03168-f003:**
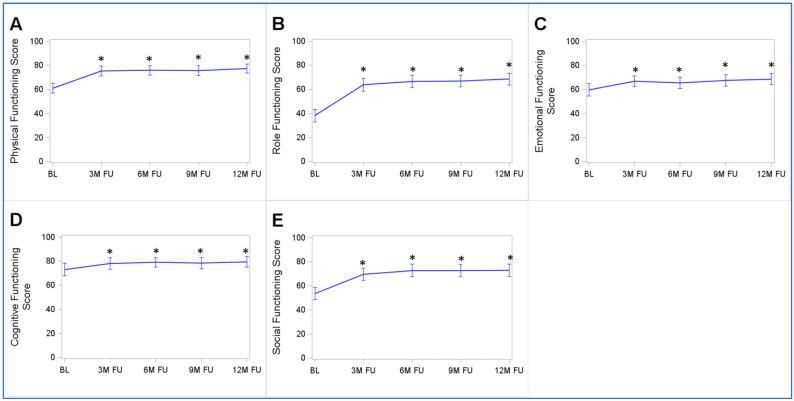
Trajectories of functional scales of the EORTC-QLQ-C30 from the baseline to 12-month follow-up: (**A**) Physical Functioning Score; (**B**) Role Functioning Score; (**C**) Emotional Functioning Score; (**D**) Cognitive Functioning Score; (**E**) Social Functioning Score. Abbreviations: BL, baseline; 3M FU, 3-month follow-up; 6M FU, 6-month follow-up; 9M FU, 9-month follow-up; 12M FU, 12-month follow-up. * *p* < 0.05 of the Wilcoxon signed-rank test comparing responses at follow-ups with those at the baseline. Note: the figure displays means with 95% confidence intervals. Only study participants with complete follow-up data were included in this analysis (*n* = 114).

**Figure 4 cancers-15-03168-f004:**
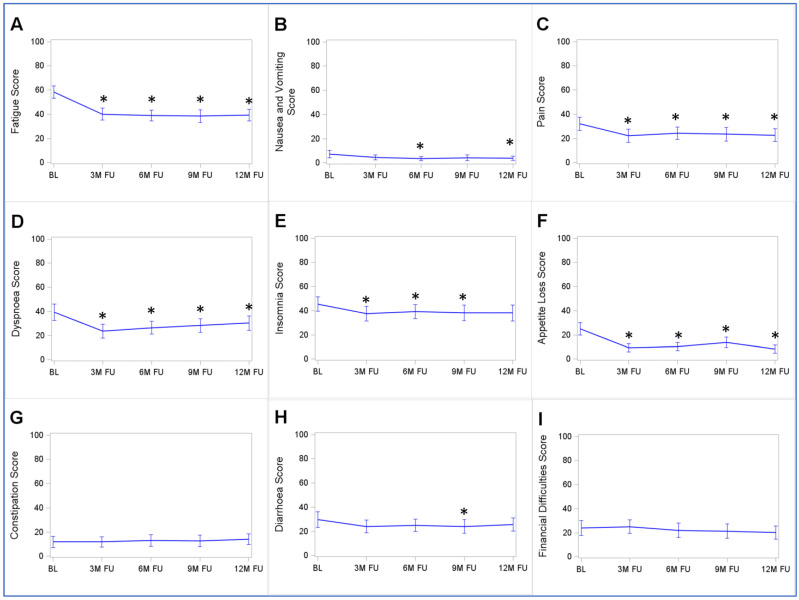
Trajectories of the symptom scales of the EORTC-QLQ-C30 from the baseline to 12-month follow-up: (**A**) Fatigue Score; (**B**) Nausea and Vomiting Score; (**C**) Pain Score; (**D**) Dyspnoea Score; (**E**) Insomnia Score; (**F**) Appetite Loss Score; (**G**) Constipation Score; (**H**) Diarrhea Score; (**I**) Financial Difficulties Score. Abbreviations: BL, baseline; 3M FU, 3-month follow-up; 6M FU, 6-month follow-up; 9M FU, 9-month follow-up; 12M FU, 12-month follow-up. * *p* < 0.05 of the Wilcoxon signed-rank test comparing responses at follow-ups with those at the baseline. Note: the figure displays means with 95% confidence intervals. Only study participants with complete follow-up data were included in this analysis (*n* = 114).

**Table 1 cancers-15-03168-t001:** Description of the study population at the baseline (N = 147).

Baseline Characteristics	N_total_	Proportion (%)	Mean (SD)
Age (years)	147		62.5 (9.9)
<65		90 (61.2)	
≥65		57 (38.8)	
Sex	147		
Female		55 (37.4)	
Male		92 (62.6)	
School education (years)	142		
9		56 (39.4)	
10–12		59 (41.5)	
13		27 (19.0)	
Months since CRC diagnosis	126		4.9 (3.9)
Cancer stage	147		
I		49 (33.3)	
II		45 (30.6)	
III		37 (25.2)	
Unknown		16 (10.9)	
Type of CRC treatment	144		
Surgery			
Yes		144 (100)	
No		0 (0)	
Chemotherapy			
Yes		76 (52.8)	
No		68 (47.2)	
Radiotherapy			
Yes		103 (71.5)	
No		41 (28.5)	
Months since CRC surgery	142		
0–1		69 (45.5)	
2–3		20 (14.1)	
4–6		23 (16.2)	
7–9		23 (16.2)	
10–12		7 (4.9)	
BMI (kg/m²)	147		27.4 (5.7)
<25		51 (34.7)	
25 to <30		60 (40.8)	
≥30		36 (24.5)	
Smoking status	129		
Never smoked		56 (43.4)	
Former smoker		57 (44.2)	
Current smoker		16 (12.4)	
Alcohol consumption ^a,b^	128		
None		26 (20.3)	
Low or moderate		94 (73.4)	
High		8 (6.3)	
Healthy physical activity ^a,c^	126		
Yes		62 (49.2)	
No		64 (50.8)	
Comorbidities			
Diabetes mellitus	128	18 (14.1)	
Hypertension	127	64 (50.4)	
History of myocardial infarction	128	5 (3.9)	
History of stroke	128	5 (3.9)	
Frailty ^d^	134		
Non-frail		19 (14.2)	
Pre-frail		89 (66.4)	
Frail		26 (19.4)	
Depression (GDS-15 scale points)	146		6 (1.7)
Anxiety (GAD-7 scale points)	144		5.4 (4.5)
Psychosocial distress (QSC-R10 scale points)	127		16 (9.6)
Social support (DUFFS-5 scale points)	132		4.4 (0.9)
Employment status	141		
Fully employed		67 (47.5)	
Part-time employed		13 (9.2)	
Retired		56 (39.7)	
Unemployed		5 (3.5)	

Abbreviations: BMI, Body Mass Index; DUFFS-5, Duke-UNC Functional Social Support Questionnaire (5-item) [29,30]; GAD-7, General Anxiety Disorder 7-item Questionnaire [20]; GDS-15, Geriatric Depression Scale 15-item Questionnaire [19]; QSC-R10, Questionnaire on Stress in Cancer Patients [18]; ^a^ during the year before the CRC diagnosis; ^b^ definitions by the WHO [31]. Definition of low or moderate alcohol consumption: women 0 to 19.99 g ethanol/day (g/d) or men 0 to 39.99 g/d; definition of high alcohol consumption: women ≥ 20 to 39.99 g/d or men ≥ 40 g/d; ^c^ physical activity was assessed with the Rapid Assessment of Physical Activity questionnaire [32]. However, we used the definition of the Healthy Lifestyle Score for healthy physical activity, which was as follows: at least 150 min of moderate-intensity or 75 min of vigorous-intensity aerobic physical activity throughout the week or an equivalent combination of moderate-intensity physical activity are needed to meet the recommendations of healthy physical activity [33]. ^d^ Assessed with the Fatigue, Resistance, Ambulation, Illnesses, and Loss of Weight (FRAIL) scale [21].

## Data Availability

The data will not be published to an open access platform. After completion of the study, interested scientists can request data use and receive pseudonymized data upon approval of this application by the principal investigator. Please contact Dr. Ben Schöttker (b.schoettker@dkfz.de).

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
