# Peer review of "Development Trajectories of Fatigue, Quality of Life, and the Ability to Work among Colorectal Cancer Patients in the First Year after Rehabilitation—First Results of the MIRANDA Study"

_cancers, 2023, doi:10.3390/cancers15123168_

Round 1

Reviewer 1 Report (Previous Reviewer 1)

Authors sufficiently answered questions/issues from the reviewer.

Author Response

We thank the reviewer for reviewing our manuscript and providing valuable insights into the topic. 

Reviewer 2 Report (Previous Reviewer 2)

To my opinion, the authors performed the revisions sufficiently. 

Author Response

We thank the reviewer for reviewing our manuscript and providing valuable insights into the topic. 

Reviewer 3 Report (Previous Reviewer 4)

As I indicated previously, the article presented is very interesting and has been improved after the indications given by the reviewers.

The article can be published. I congratulate the researchers for the great work they have done.

Author Response

We thank the reviewer for reviewing our manuscript and providing valuable insights into the topic. 

This manuscript is a resubmission of an earlier submission. The following is a list of the peer review reports and author responses from that submission.

Round 1

Reviewer 1 Report

11.  Patients with cancer including CRC are at increased risk of disease and treatment related symptoms such as fatigue, pain etc.. that effects quality of life. Many patients may benefit from an access to rehabilitation services, thus MIRANDA study is of high importance. This is preliminary analysis of first 147 patients from the planned enrollment of 1000 to MIRANDA study. What rehabilitation services were/are offered/available to these patients? The manuscript may benefit from better description of rehabilitation process patient in the study received.

22. Important: of 144 patients 114 had data for all assessment. How the characteristics of patients who did not reach 12M follow-up differ from those who reached 12M assessment. What were the reasons for drop-out.

33. weighted Kappa statistic and Table S1: The information about Kappa statistics that is currently placed in 2.2.3 (methods outcome of interest ability to work) belongs 1) to statistical method (the methodology part of information) and the results (the result part of information).  Also, within the supplementary materials Table S1has title “Weighted Kappa statistic of the Work Ability” However, this reviewer is not able to see Kappa statistics anywhere within the table.  Consider labeling the Table S1 properly.

44. For easier reading, for results in sections 3.2, 3.3 and 3.4 reemphasize within the main text that these are assessments at baseline.

53. Figure 2,3,4: what is are listed p-values evaluating, please specify within the footnote.

64.    Power misspelled as “popwer” line 423

Just minor problems e.g. power misspelled as “popwer” line 423

Reviewer 2 Report

General comments:

The present manuscript deals with an important topic, which concerns patients suffering from colorectal cancer.  The authors performed a study to evaluate inter-correlation of the parameters fatigue, quality of life, and the ability to work and their trajectories in the first year after in-patient rehabilitation in a large ongoing study. The present manuscript is well written. It allows important insights in this topic. The methods were appropriate, and the results as well as the discussion are written in a clear manner. Nevertheless, there seem to be limitations, which have to be addressed.

Special comments:

Please correct the tipping error.

Methods:

- Comorbidities: Were there patients with notable comorbidities like diabetes mellitus, cardiac comorbidity? What were your exclusion criteria?

- Why did you perform a retrospective study? Furthermore, there is no control group. Please discuss this limitation

- Page 3, line 119-134: …”Every three months in the first year and then after 3, 5, 7, and 10 years, study participants are again asked to fill out self-administered questionnaires containing the following validated endpoint scales:”…:

You used a lot of scales? My experiences in large databases is that this can cause a many missing data. Furthermore, the individual participanti is not completely anonymous and by himself when filling out the questionnaires, and is therefore more likely to give socially accepted answers. Please discuss these factors

Results:

Page 5, Table 1: you included patients with different cancer stages. Might this influence your results?

Reviewer 3 Report

The authors report the results of 9 scores relating to QoL, fatigue and perceived ability or inability to work in a small, inhomogen sample of 147 patients with colorectal cancer having benefited from an inpatient rehabilitation at various time points during this first year after start if treatment.

The only report the median values, no range, no influence of disease stage, time of treatment... So the information "gained" is minimized.

The beneficial effect of rehabilitation, an effect on the potential return to work in such a small sample with near half of patients already in retirement age cannot be assessed like this

Reviewer 4 Report

The work presented by the researchers is very interesting, but as they comment, the sample size is small and cannot be extended to the rest of the population. The methodology and results presented are clearly and concisely written.

It would be necessary to wait for the number of patients to be increased to see if the results hold up.  In addition, the age of the participants is important with regard to the ability to work.  The following bibliographic citation may help in the discussion of this article:

*Arndt V, Merx H, Stegmaier C, Ziegler H, Brenner H. Quality of life in patients with colorectal cancer 1 year after diagnosis compared with the general population: a population-based study. J Clin Oncol. 2004 Dec 1;22(23):4829-36. doi: 10.1200/JCO.2004.02.018. Erratum in: J Clin Oncol. 2005 Jan 1;23(1):248. 

Bibliographic references 31, 32, 42 are not correctly cited and should be corrected.

The English Language is correct